# An Exosome-Rich Conditioned Medium from Human Amniotic Membrane Stem Cells Facilitates Wound Healing via Increased Reepithelization, Collagen Synthesis, and Angiogenesis

**DOI:** 10.3390/cells12232698

**Published:** 2023-11-24

**Authors:** Chan Ho Noh, Sangryong Park, Hye-Rim Seong, Ah-young Lee, Khan-Erdene Tsolmon, Dongho Geum, Soon-Cheol Hong, Tae Myoung Kim, Ehn-Kyoung Choi, Yun-Bae Kim

**Affiliations:** 1College of Veterinary Medicine, Chungbuk National University, Cheongju 28644, Republic of Korea; chnoh@designedcells.com (C.H.N.);; 2Central Research Institute, Designed Cells Co., Ltd., Cheongju 28576, Republic of Koreatmkim@designedcells.com (T.M.K.); ekchoi@designedcells.com (E.-K.C.); 3Department of Biomedical Science, Korea University College of Medicine, Seoul 02841, Republic of Korea; 4Department of Obstetrics and Gynecology, Korea University College of Medicine, Seoul 02841, Republic of Korea

**Keywords:** amniotic membrane stem cell, exosome-rich conditioned medium (ERCM), growth factor, neurotrophic factor, wound healing, keratinocyte proliferation, collagen synthesis, angiogenesis

## Abstract

Tissue regeneration is an essential requirement for wound healing and recovery of organs’ function. It has been demonstrated that wound healing can be facilitated by activating paracrine signaling mediated by exosomes secreted from stem cells, since exosomes deliver many functional molecules including growth factors (GFs) and neurotrophic factors (NFs) effective for tissue regeneration. In this study, an exosome-rich conditioned medium (ERCM) was collected from human amniotic membrane stem cells (AMSCs) by cultivating the cells under a low oxygen tension (2% O_2_ and 5% CO_2_). The contents of GFs and NFs including keratinocyte growth factor, epidermal growth factor, fibroblast growth factor 1, transforming growth factor–β, and vascular endothelial growth factor responsible for skin regeneration were much higher (10–30 folds) in the ERCM than in normal conditioned medium (NCM). In was found that CM–DiI-labeled exosomes readily entered keratinocytes and fibroblasts, and that ERCM not only facilitated the proliferation of keratinocytes in normal condition, but also protected against H_2_O_2_ cytotoxicity. In cell-migration assay, the scratch wound in keratinocyte culture dish was rapidly closed by treatment with ERCM. Such wound-healing effects of ERCM were confirmed in a rat whole skin-excision model: i.e., the wound closure was significantly accelerated, remaining minimal crusts, by topical application of ERCM solution (4 × 10^9^ exosome particles/100 μL) at 4-day intervals. In the wounded skin, the deposition of collagens was enhanced by treatment with ERCM, which was supported by the increased production of collagen-1 and collagen-3. In addition, enhanced angiogenesis in ERCM-treated wounds was confirmed by increased von Willebrand factor (vWF)-positive endothelial cells. The results indicate that ERCM from AMSCs with high concentrations of GFs and NFs improves wound healing through tissue regeneration not only by facilitating keratinocyte proliferation for skin repair, but also activating fibroblasts for extracellular matrix production, in addition to the regulation of angiogenesis and scar tissue formation.

## 1. Introduction

Wound healing is a dynamic process encompassing keratinocyte and fibroblast migration, proliferation, reepithelization, matrix synthesis and remodeling, and neovascularization [1,2]. In gross and microscopic findings, defective skins are repaired by the formation of granulation tissue composed of collagen matrix and new vessels [2]. The formed granulation tissue also provides the substrate for the reestablishment of neoepidermis by keratinocyte proliferation and migration. During these processes, wound contraction contributes to wound closure [3,4].

Wound contraction undergoes via biphasic processes. The first phase is immediate and tension-free. The second phase is characterized by increased contraction mediated by α-smooth muscle actin (α-SMA)-containing myofibroblasts [5]. Mechanical force is generated by myofibroblasts and transmitted into the surrounding extracellular matrix (ECM), leading to matrix reorganization and wound contraction.

In more detailed skin injury and healing processes, following skin damage, hemo-stasis and inflammation occur, blood vessels constrict, and fibrin and platelets aggregate, forming clots [6]. These clots release cytokines and GFs that initiate an inflammatory response. Neutrophils, monocytes, fibroblasts, and endothelial cells at the wound site secrete ECM, providing a scaffold in which tissue-regenerating cells can be activated and proliferate more appropriately. Inflammatory mediators accumulate, blood vessels around the wound expand, and cells infiltrate to remove invading microorganisms and cellular debris [7,8].

Fibroblasts, present during the emergence of new blood vessels, proliferate and invade the thrombus to form contractile granulation tissue [9,10]. At this time, as the wounds matures, major collagens make up the ECM. More than 20 different types of collagen have been identified from humans, mainly types 1, 2, and 3. Collagen is one of the popular materials in protein-based scaffolds, and collagen–1 and collagen–3 play an important role in wound healing. During wound healing, the components of the ECM show certain changes: i.e., collagen–3 produced in the proliferative phase is replaced by stronger collagen–1 as the tissue matures [11,12]. Remodeling, degradation, and synthesis of the ECM will strengthen the tensile strength and restore the normal tissue structure, in which the nurturing tissue will gradually rise, the hair follicles and blood vessels will be formed, and the collagen fibers will also gradually increase [13].

It has been known that stem cells play a central role in the wound healing through paracrine effects. Notably, the paracrine effects are mediated by functional molecules including proteins and microRNAs (miRNAs) released from stem cells [14,15]. Inter-estingly, the functional molecules are delivered in forms of extracellular vesicles (EVs) called exosomes, by which healing efficiency is increased. Exosomes are membrane lipid vesicles with diameters of 30–250 nm and have attracted a lot of attention in the field of skin recovery and regeneration. Indeed, stem cell-derived exosomes are known to accelerate wound closure and promote wound healing [16]. As underlying mechanisms, stem cell exosomes from adipose-derived stem cells (ADSCs) and umbilical cord blood stem cells (UCBSCs) modulate inflammatory responses, accelerate angiogenesis, increase keratinocyte and fibroblast migration and proliferation, and activate fibroblasts to synthesize collagen and elastin fibers [17,18]. Interestingly, acellular amniotic membrane incorporating exosomes from ADSCs further potentiated the ADSC exosomes’ promoting effects on the diabetic wound healing [19].

Recently, we attained an ERCM from human AMSCs by cultivating at a hypoxic oxygen (2% O_2_) tension [20]. In our previous study, AMSCs did not cause immune rejection and inflammation, since they do not express major histocompatibility class (MHC) type II [21]. It was confirmed that the ERCM contained tens or hundreds of functional molecules compared with a NCM collected at a normoxic (20% O_2_) culture condition. Therefore, in the present study, we assessed whether the AMSC exosomes enter keratinocytes and fibroblasts, facilitate keratinocyte proliferation and fibroblast activation, enhance angiogenesis, and thereby enhance wound healing in vitro and in vivo.

## 2. Materials and Methods

### 2.1. Preparation of AMSCs

AMSCs were collected under Good Manufacturing Practice conditions (Central Research Institute of Designed Cells Co., Ltd., Cheongju, Republic of Korea) as previously described [20,21]. In brief, amniotic membranes were digested with collagenase I. After removal of red blood cells, the remaining cells were suspended in Designed Cells-Exclusive Medium (DCEM; Designed Cells Co., Ltd.) supplemented with 5% fetal bovine serum (FBS), 100 U/mL penicillin, and 100 mg/mL streptomycin (Invitrogen, Carlsbad, CA, USA). Cultures were maintained under 5% CO_2_ at 37 °C in culture flask. The prepared amniotic stem cells were analyzed for their stem cell markers in a fluorescence-activated single cell sorting (FACS). The cells were confirmed to be mesenchymal stem cells (MSCs) [21].

### 2.2. Preparation of ERCM and Characterization

The separated AMSCs were dispersed in serum-free DCEM (1 × 10^8^/mL) in Hyper flask (Nunc, Rochester, NY, USA) and cultivated under normoxic oxygen (20% O_2_, 5% CO_2_) or hypoxic oxygen (2% O_2_, 5% CO_2_) tensions at 37 °C for 3 days [20]. The medium was vacuum-filtered through a PES membrane (0.22 μm) (Corning, Glendale, CA, USA). The conditioned medium was concentrated 30 times using Vivaflow-200 (Sartorius, Hannover, Germany).

For transmission electron microscopic (TEM) observation, exosomes loaded on Formvar-coated copper grids were fixed with paraformaldehyde and glutaraldehyde, and counterstained with uranyl acetate to show the images of exosomes. Using JEM-plus TEM instrument (Jeol, Tokyo, Japan) equipment with Oneview camera (Gatan, Berwyn, IL, USA), the images were attained at 30,000 magnifications [22].

To analyze size distribution of exosomes, nanoparticle-tracking analysis (NTA) were performed using Nanosight NS 300 equipped with v3.2.16 analytical software (Malvern Panalytical, Malvern, UK). Purified exosomes were adjusted to 10^8^ particles/mL (in accordance with the manufacturer’s recommendations) in PBS. The camera level was adjusted until the particle signal saturation did not exceed 20% and all particles were clearly visible, 5 images were captured in 60 s, the size and particle concentration were analyzed [23,24].

Western blot analysis of CD9, CD63, and CD81 from isolated exosomes was performed using protein DC assay kits (Bio-Rad Laboratories, Hercules, CA, USA) [20]. An aliquot of NCM or ERCM was denatured using denaturation buffer, and then resolved via 12% SDS-PAGE. Resolved proteins were transferred onto Immobilon-P PVDF membrane and reacted with primary antibodies for CD9, CD63 or CD81 (1:1000; Abcam, Cambridge, UK) overnight at 4 °C, followed by incubation with horseradish peroxidase (HRP)-conjugated secondary anti-mouse antibody (1:2000; Abcam, Cambridge, UK) at room temperature. After washing, the signal was recorded using WestFemto maximum sensitivity substrate kit under Bio-Rad ChemiDoc Imager (Bio-Rad Laboratories, Hercules, CA, USA), as shown in Appendix A.

Enzyme-linked immunosorbent assay (ELISA) was conducted to analyze functional molecules, that is, GFs and NFs, related to wound healing in NCM and ERCM [20]. ELISA kits for keratinocyte growth factor (KGF) (ab183362; Abcam, Cambridge, UK), epidermal growth factor (EGF) (K0331115; Komabiotech, Seoul, Republic of Korea), fibroblast growth factor 1 (FGF1) (ab219636; Abcam, Cambridge, UK), transforming growth factor–β (TGF–β) (ab100647; Abcam, Cambridge, UK), and vascular endothelial growth factor (VEGF) (ab100662; Abcam, Cambridge, UK) were used according to the manufacturer’s instructions. Briefly, NCM or ERCM was put into the ELISA wells and incubated at room temperature. After washing 3–4 times, the primary antibodies were added, and reacted at room temperature. Following incubation with secondary antibody at room temperature, color-developing substrate was applied for 10–30 min. After treatment with a stop solution, the absorbance was measured at 450 nm.

### 2.3. Exosome Uptake, Cell Proliferation and Protection Assay

Exosomes in ERCM obtained from AMSCs were labeled with red CM-DiI membrane dye (C7000; Invitrogen, Carlsbad, CA, USA) and prepared at a concentration of 4 × 10^10^ particles/mL [25,26]. In brief, freeze-dried ERCM was dissolved in PBS (4 × 10^10^ particles/mL). The prepared exosomes were treated with CM-DiI membrane dye diluted (1:1000) with a stock solution prepared according to the manufacturer’s instructions, and incubated for 10 min at 4 °C. The labeled exosomes were ultracentrifuged at 100,000× *g* for 4 h (Beckman, Brea, CA, USA), and the pellets were resuspended in PBS to make 50 μL/mL.

HaCaT (a human keratinocyte cell line; Cell Lines Service, Heidelberg, Germany) and 3T3–L1 (a mouse fibroblast cell line; ATCC CL–173, Manassas, BA, USA) cells were cultivated in Dulbecco’s Modified Eagle’s Medium (DMEM; Biowest, Kansas City, MO, USA) supplemented with 10% fetal bovine serum (FBS), 1% penicillin/streptomycin, and 1% L-glutamine (Gibco Life Technologies, Grand Island, NY, USA). The HaCaT and 3T3–L1 cells were seeded on 8-well chamber slides (NUNC C7182; Thermo Fisher Scientific, Waltham, MA, USA) at 1 × 10^5^/mL. After 24 h, cells were damaged with 200 µM H_2_O_2_, and then incubated with CM-DiI-labeled exosomes (50 μL/mL) for 4 h at 37 °C [20]. The cells were fixed with 4% paraformaldehyde, and treated with 0.1% Triton X–100 (Thermo Fisher Scientific, Waltham, MA, USA). After blocking with 1% bovine serum albumin (BSA) for 1 h, the cells were immunostained with anti-α-tubulin antibody (1:1000, ab7291; Abcam, Cambridge, UK) for 2 h at 37 °C, followed by goat anti-mouse IgG Alexa FluorTM 488 (1:500, Invitrogen, Carlsbad, CA, USA) for 1 h at room temperature. The cell nuclei were stained with DAPI (Thermo Fisher Scientific, Waltham, MA, USA), and examined under a microscope (BX51; Olympus, Tokyo, Japan) [25,26].

HaCaT cells (1 × 10^4^/mL) were seeded in a 96-well plate. In order to assess the cell-proliferating activity of ERCM, the cells were treated with ERCM (1–30 μL/mL). To evaluate the cytoprotective activity of ERCM against oxidative stress, the cells were exposed to 200 μM H_2_O_2_ and treated with ERCM [20]. After 24-h culture at 37 °C, the cell viability was quantified using Cell Counting Kit–8 (CCK–8; Dojindo, Kumamoto, Japan). CCK-8 assay was carried out by adding 10 µL of CCK–8 reagent into each cell culture well. After additional 2-h incubation, the absorbance at 450 nm was measured using a microplate reader (Bio-Rad Laboratories Inc., Hercules, CA, USA).

For the in vitro wound-healing (scratched cell-migration) assay, HaCaT cells (1 × 10^5^/mL) were seeded in a 12-well plate for about 24 h. When 90% confluence was achieved, a uniform scratch was made with a sterile pipette tip. Each well was washed with PBS, supplemented with new culture medium containing each concentration of ERCM, and then cultivated for 24 h. Images of each scratch were observed microscopically, and captured immediately and 24 h after scratching.

### 2.4. Collagen Synthesis in 3T3–L1 Cells

For western blot analysis of collagens, 3T3–L1 cells (1 × 10^4^/mL) were seeded in a 96-well plate and treated with ERCM (1–30 μL/mL). After 24-h culture at 37 °C, the cells were collected and homogenized in 10 volumes of RIPA buffer solution (Pierce, Rockford, IL, USA). Proteins were quantified using a BCA protein assay kit (Pierce, Rockford, IL, USA). Proteins were denatured in 0.5 M Tris-HCl buffer (pH 6.8) containing 10% SDS and 10% ammonium persulfate, separated by 10% SDS-PAGE, and transferred to a nitrocellulose membrane in 25 mM Tris buffer containing 20% methanol, 1% SDS, and 192 mM glycine. After blocking with 3% skim milk, the membrane was incubated with a collagen–1 (1:1000, PA5-29569; Invitrogen, Carlsbad, CA, USA) or collagen–3 (1:1000, PA5-27828; Invitrogen, Carlsbad, CA, USA) overnight at 4 °C, followed by a secondary goat anti-rabbit IgG conjugated with HRP (1:1000, 7074; Cell Signaling Technology, Danvers, MA, USA) for 1 h at room temperature. The membrane was then developed using an enhanced chemiluminescence solution (ATTO-TEC GmbH, Siegen, Germany). The band densities were measured using ImageJ software (version v1.54d; National Institutes of Health, Bethesda, MD, USA).

### 2.5. Whole-Thickness Wound Model and Wound Closure

Male Sprague-Dawley rats (7 weeks old, weighing 200–250 g) were purchased from DBL (Eumseong, Republic of Korea). The animals were housed at a standard room with a constant temperature (23 ± 2 °C), relative humidity of 55 ± 10%, and 12-h light/dark cycle, and given commercial rodent chow and purified water ad libitum.

The fur on the dorsal skin of the rats was shaved with a clipper prior to preparing the wound. Rats (*n* = 10/group) were lightly anesthetized with isoflurane, and then a full-thickness excision wound was made using a sterile 20-mm (in diameter) biopsy punch. ERCM (4 × 10^9^ exosome particles/100 μL) or Saline solution was applied over the wound area 4 times on days 0, 4, 8, and 12. As a reference material, 3 mg Fucidin^®^ ointment (100 μg as fusidate sodium; Donghwa Pharm, Seoul, Republic of Korea) was applied at the same schedule.

The wound areas were observed on days 0, 4, 8, 12, 16, and 19, photographed, and quantified using ImageJ software (National Institutes of Health, Bethesda, MD, USA). The rate of wound closure was expressed as the ratio of wound remained compared to the whole area on day 0.

### 2.6. Microscopic Observation and Myofibroblast Analysis

Following tissue-processing procedures, formalin-fixed and paraffin-embedded skin tissues were stained with hematoxylin-eosin and Masson’s trichrome to observe overall structure and fibers, respectively. All sections were counterstained with Mayer’s hematoxylin, and examined under a high-power optical microscope (Carl Zeiss, Jena, Germany).

In order to confirm myofibroblast differentiation, α-SMA was immunostained. The tissue sections were incubated with a primary antibody specific for α-SMA (1:300, ab7817; Abcam, Cambridge, UK) overnight at 4 °C. After washing with PBS, the sections were incubated with a secondary antibody conjugated with Fluor–488 (1:1000, ab150077; Abcam, Cambridge, UK) for 1 h at room temperature. Images were taken with a fluorescence microscope (BX53; Olympus, Tokyo, Japan), and the area was quantified using ImageJ software (National Institutes of Health, Bethesda, MD, USA).

### 2.7. Collagen and Microvessel Analysis

In order to clearly show the collagen synthesis in fibroblasts, collagen–1 and collagen-3 were immunostained. The tissue sections were incubated with primary antibodies specific for collagen–1 (1:1000, PA5-29569; Invitrogen, Carlsbad, CA, USA) or collagen–3 (1:1000, PA5-27828; Invitrogen, Carlsbad, CA, USA) overnight at 4 °C. After washing with PBS, the sections were incubated with secondary antibodies for 2 h. The tissues were color-developed with 3,3-diaminobenzidine tetrahydrochloride (DAB; Novus Biologicals, Centennial, CO, USA) for 1–2 min. All sections were counterstained with Mayer’s hematoxylin, and examined under a high-power optical microscope (Carl Zeiss, Jena, Germany).

For western blot analysis of collagens, the wounded skins were excised, freeze-clamped in liquid nitrogen, and stored at −80 °C until use. Frozen tissue samples were homogenized in 10 volumes of RIPA buffer solution (Pierce, Rockford, IL, USA), and the quantities of collagen–1 and collagen-3 were obtained from the band densities described above.

In order to confirm angiogenesis, vWF was immunostained. The tissue sections were exposed to proteinase K, and incubated with a primary antibody specific for vWF (1:300, ab287962; Abcam, Cambridge, UK) overnight at 4 °C. After washing with PBS, the sections were treated with a biotinylated secondary antibody (Vectastain Elite ABC kit; Vector, Burlingame, CA, USA) for 60 min. The tissues were color-developed with DAB (Vector, Burlingame, CA, USA) for 1–2 min. All sections were counterstained with Harris’s hematoxylin, and examined under a high-power optical microscope (Carl Zeiss, Jena, Germany). Images were taken with a fluorescence microscope (Olympus, Tokyo, Japan), and the area was quantified using ImageJ software (National Institutes of Health, Bethesda, MD, USA).

### 2.8. Statistical Analysis

Data are presented as mean ± standard deviation. Statistical analysis was performed with SPSS version 26.0 program (SPSS Inc., Chicago, IL, USA). Differences among groups were analyzed with one-way ANOVA, followed by Tukey’s HSD at a level of *p* < 0.05.

## 3. Results

### 3.1. Characteristics of Exosomes from AMSCs

From TEM analysis, typical exosome structures of homogeneous, spherical, and membrane-bound vesicles were observed, in which the size of the exosome particles was confirmed to be smaller than 100 nm (Figure 1A). NTA revealed the size distribution of exosomes, wherein the major peak was found to be 72 nm (Figure 1B). The particle numbers of exosomes in NCM and ERCM were calculated to 8.8 × 10^8^/mL and 4.5 × 10^10^/mL, respectively.

The contents of exosomes in NCM and ERCM were measured by western blotting CD9, CD63, and CD81 markers. In comparison with NCM obtained in a normoxic condition (20% O_2_), levels of all the 3 markers in ERCM collected after a hypoxic culture (2% O_2_) were much higher (Figure 1C). In parallel with the exosome markers, the concentrations of GFs and NFs including KGF, EGF, FGF1, TGF-β, and VEGF were very high in ERCM, reaching 50 times those of NCM (Figure 1D).

### 3.2. Exosome Uptake in HaCaT and 3T3–L1 Cells

It was confirmed that the CM–DiI-labeled AMSC exosomes readily penetrated both the normal and H_2_O_2_-treated HaCaT keratinocytes and 3T3–L1 fibroblasts, although more exosomes were found in the damaged cells (Figure 2).

### 3.3. Proliferative and Protective Activities in HaCaT Cells

In a normal culture condition, ERCM significantly increased the HaCaT keratinocyte proliferation at concentrations of 6–30 μL/mL, reaching up to 162% for 24 h (Figure 3A). As an oxidative stress, HaCaT cells exposed to 200 μM H_2_O_2_ for 24 h resulted in 35% death (Figure 3B). However, simultaneous treatment with the ERCM near-fully rescued the keratinocytes from a low concentration of 1 μg/mL, and furthermore facilitated their proliferation at a high concentration (30 μg/mL).

In a model of in vitro wound-healing, ERCM facilitated the HaCaT cell migration in a concentration-dependent manner (Figure 3C,D). Although the scratch wound was closed by 53% without treatment, it was significantly facilitated by treatment with 6–30 μg/mL of ERCM, reaching 84–95%.

### 3.4. Collagen Synthesis in 3T3-L1 Cells

It was confirmed that the synthesis of collagen–1 protein was markedly facilitated by ERCM in a concentration-dependent manner in 3T3-L1 fibroblasts, although collagen–3 tended to increase to some extant (Figure 4).

### 3.5. Wound Healing In Vivo

The skin wound applied with Saline solution was gradually closed, reaching 90% 19 days after whole-thickness excision (Figure 5A,B). The wound healing was markedly accelerated by treatment with the ERCM: i.e., the closure rate was much higher at early phase, leading to 90% on day 12, without prominent scar tissue formation. A partial effect was also achieved with fusidate sodium.

### 3.6. Microscopic Findings and Myofibroblast Differentiation

The histological structures of the regenerated dermis were analyzed on day 19 (Figure 6A). In hematoxylin-eosin stained findings, wide regenerating tissues were observed. Notably, the wound treated with ERCM was narrower than the saline-treated wound, and hair follicles were formed in the restored tissue. In the trichrome-stained findings, the facilitated wound closure with minimal scar tissues was more clearly observed when treated with ERCM, and fusidate exhibited a partial effect.

In the immunohistochemical (IHC) analysis, the α-SMA-positive myofibroblasts increased in the wound were significantly attenuated by treatment with ERCM (Figure 6A,B). Fusidate also tended to decrease the α-SMA-positive cells to some extent.

### 3.7. Collagen Synthesis and Neovascularization

In the IHC analysis of collagen–1 and collagen–3, the contents of the collagens were upregulated by treatment with ERCM (Figure 7A). Such increased production of collagens was confirmed by western blot analysis (Figure 7B,C). Separately, fusidate tended to increase only the collagen–3 production.

As confirmed by vWF-positive endothelial cells, angiogenesis in the regenerating tissue was greatly promoted by treatment with ERCM (Figure 7A,D). In comparison, fusidate exhibited a negligible effect.

## 4. Discussion

Wound healing proceeds through an inflammatory phase, proliferation of pre-existing cells, ECM repair by collagen deposition, and post-differentiation processes [27,28,29]. During this process, various cell types including keratinocytes, fibroblasts, endothelial cells, macrophages, and platelets, are regulated, of which migration, invasion, proliferation, and differentiation are involved in inflammatory responses, new tissue formation, and ultimately wound closure. This complex process is executed and coordinated by a highly complicated signaling network involving numerous GFs as well as cytokines and chemokines [27,28,29]. Hence, the topic of wound healing has attracted intensive research to understand its pathophysiology and develop new treatment strategies of a variety of diseases.

In wound healing, small EVs are considered to have the capacity influencing key biological processes for skin regeneration [30]. Interestingly, MSCs play a key role in tissue regeneration: i.e., exosomes secreted from stem cells may contribute to paracrine signaling [31]. Notably, it has been well documented that hypoxia plays an important role in maintaining stem cell plasticity and proliferation, and especially, MSCs proliferate faster and secret more EVs via enhanced gene expression and metabolism in hypoxic conditions than in normoxic environment, indicating that they are activated in the tissue with oxygen tension lower than in the air [32,33,34]. Similarly, we attained ERCM containing more exosomes and higher concentrations of GFs/NFs than in NCM via cultivation of AMSCs in a hypoxia condition.

We demonstrated that the local application of AMSC ERCM into rat whole-skin-excision wound exerted prominent repairing effects, as confirmed by more rapid closure, enhanced collagen deposition, higher rates of reepithelization and new microvessel formation, as well as less scar tissue formation. Notably, in a previous study, similar reduced scar formation was achieved with ADSC exosomes [17]. Moreover, a human acellular amniotic membrane (hAAM) provided ADSC exosomes with good environment for regulating inflammation, stimulating vascularization, and promoting the production of ERM in the diabetic skin wound [19]. Although the authors showed that the hAAM without cell ingredients could be a good scaffold for clinical applications, we demonstrated that AMSC ERCM can be used without scaffold for the enhanced wound healing.

Such effects of ERCM on the enhanced wound repair could be supported by the activity on cell proliferation and migration. It is interest to note that the exosome particles readily penetrated both the normal and H_2_O_2_-exposed keratinocytes and fibroblasts, wherein more exosomes were found in the damaged cells. Particularly, the particle size of AMSC exosomes (mean 72 nm) was much smaller than those from other MSCs such as ADSCs (mean 220 nm) and UCBSCs (mean 120 nm) [35,36]. Such a small size may be advantageous for passing body membranes including the skin and blood–brain barrier (BBB). In addition, ERCM markedly facilitated keratinocyte proliferation, protected against oxidative injury, and induced migration into a scratch wound in culture systems [18]. Interestingly, the cell-proliferative and -protective effects were observed at very low concentrations (≥3–6 μL/mL), especially displaying near-full recovery from H_2_O_2_ cytotoxicity at 1 μL/mL. The results suggest that exosomes providing GFs and NFs may directly protect keratinocytes and/or affect keratinocytes to evoke their regenerative responses, thereby accelerating proliferation and overcoming cytotoxicity. Indeed, high concentrations of KGF, EFG, and VEGF were detected in ERCM in the ELISA. Actually, it was reported that stem cells highly expressing GFs and NFs protected brain injury in a middle cerebral artery occlusion (MCAO) model [37], and that VEGF enhanced antioxidative capacity by up-regulating antioxidant enzymes [38]. Therefore, the decreased cell and tissue injuries may be in part mediated by antioxidative activity of GFs/NFs released from ERCM [39,40].

Exosomes up-regulate protein expression related to angiogenesis, reepithelization, and ECM remodeling in wounded tissues [27,28,29]. Angiogenesis is known to play a central role in wound repair, wherein VEGF is the major GFs involved. The initial inflammation is a key phase in the wound healing process, which partially triggers the angiogenic response by producing high levels of VEGF [31,41].

In addition to KGF and EGF, the epithelial cell-specific GFs, very high concentrations of FGF1 and TGF–β related to fibroblast activation and proliferation were also observed in ERCM [20,42]. FGF1 is the basic GF for ECM production through collagen synthesis. Therefore, it is believed that the increased synthesis of collagen–1 and collagen–3 and their deposition in the wounded tissue may be partially induced by FGF1 from ERCM. The up-regulation of VEGF expression increases the production of TGF–β, which is well known to play important roles in wound healing and scar prevention [43]. Actually, TGF–β was found to have important implications in all phases of wound healing, with recent dramatic findings implicating treatment options for early wound control. From the effects of TGF–β on monocytes, fibroblasts, endothelial cells, and keratinocytes, it was confirmed that this growth factor has pivotal roles in each phase of wound healing [43]. In general, TGF–β is released from degranulating platelets and secreted by all of the major cell types participating in the repair process, including lymphocytes, macrophages, endothelial cells, smooth muscle cells, epithelial cells, and fibroblasts [44,45]. Importantly, ERCM contained high levels of TGF–β governed the early phase of wound repair, leading to facilitated wound closure as well as decreased inflammatory complications.

In late proliferative phase, ECM production, angiogenesis, and reepithelization are major phenomena. In the maturation phase, formation of myofibroblasts for wound contraction is commonly observed [46]. Importantly, it is well known that TGF–β1 induces myofibroblast differentiation. Indeed, it was demonstrated that impaired wound contraction was associated with a reduction of α-SMA-positive myofibroblasts in granulation tissue [6]. In the present study, it is believed that the high concentration of TGF–β in ERCM led to a rapid wound contraction. In addition to early role, TGF–β is also involved in the inflammation, angiogenesis, reepithelization, and connective tissue regeneration. That is, TGF–β regulates angiogenic VEGF expression, collagen production (particularly collagen–1 and collagen–3) composing ECM, and remodeling of wounds affecting scar tissue formation [47,48]. Thus, the beneficial effects, increased wound contraction and repair, might came in part from TGF–β in ERCM.

In microscopic IHC findings, increased collagen deposition was observed in the deep wound treated with ERCM. Such findings were confirmed by the western blot analysis of collagen–1 and collagen–3, indicative of the accelerated wound healing showing earlier closure of the open injury [49,50]. The histological analysis confirmed that the wound healing entered the remodeling stage with changes in collagen types in ECM [51]. It was demonstrated that exosomes from ADSCs inhibited trans-differentiation of fibroblasts-to-myofibroblasts and hypertrophic scar formation via miR–192-5p – IL–17RA – Smad axis [35]. In the present study, ERCM markedly decreased the α-SMA-positive cells in the wound, compared with a mild effect of fusidate, an antibiotic eliminating bacterial contamination. Indeed, the surface tissue crust showed inverted contraction in the wound applied with saline. Interestingly, however, the crust formation was minimal when treated with ERCM. So, it is suggested that ERCM regulated the trans-differentiation of fibroblasts at the late stage of wound closure, leading to more smooth surface healing and remaining minimal scar tissue [52].

## 5. Conclusions

Collectively, our research findings demonstrate that ERCM from AMSCs can promote skin wound healing and reduce scar tissue formation by facilitating proliferation of keratinocytes and optimizing the properties of fibroblasts. Although additional follow-up studies to clarify underlying mechanisms, it is suggested that ERCM containing a large amount of GFs and NFs could be a good candidate for clinical trials to manage diverse acute and chronic skin lesions.

## Figures and Tables

**Figure 1 cells-12-02698-f001:**
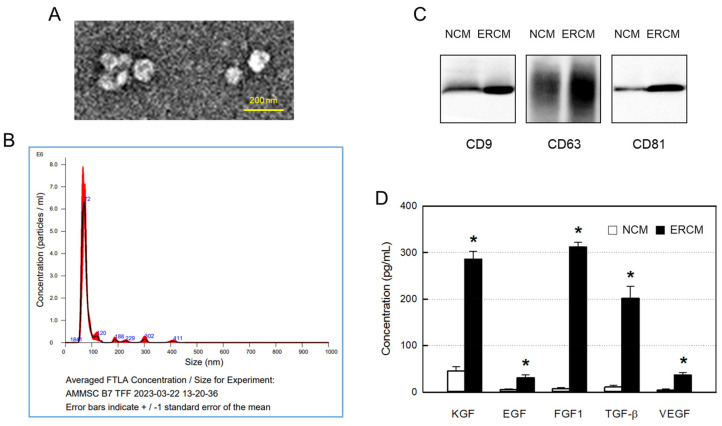
Isolation and characterization of exosomes from amniotic membrane stem cells (AMSCs). (**A**) Representative transmission electron microscopic findings of exosomes. (**B**) Particle size distribution of AMSC exosomes analyzed by a Nanoparticle-Tracking Analysis system. Red line: calibration curve for quantification, blue number: size of each peak. (**C**) Western blot analysis of CD9-, CD63-, and CD81-positive exosomes in normal conditioned medium (NCM) and exosome-rich conditioned medium (ERCM). (**D**) Concentrations of growth factors (GFs) and neurotrophic factors (NFs) in NCM (white bars) and ERCM (black bars). KGF: keratinocyte growth factor, EGF: epidermal growth factor, FGF1: fibroblast growth factor 1, TGF–β: transforming growth factor–β, VEGF: vascular endothelial growth factor. * Significantly different from NCM (*p* < 0.05).

**Figure 2 cells-12-02698-f002:**
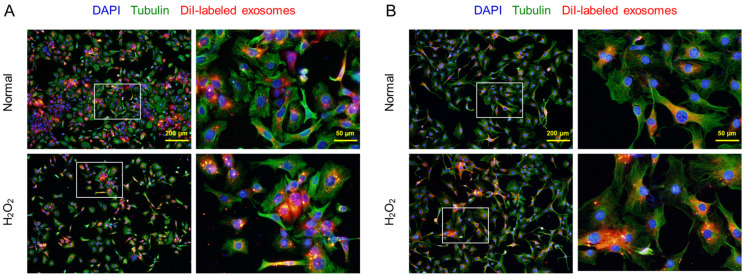
Penetration of CM-DiI-labeled AMSC exosomes into HaCaT keratinocytes (**A**) and 3T3–L1 fibroblasts (**B**). The cells (1 × 10^5^/mL, *n* = 5/group) were treated with 200 µM H_2_O_2_ or its vehicle (Normal), and incubated with labeled exosomes (50 μL/mL) for 4 h. Inset: part for each enlarged image (right side).

**Figure 3 cells-12-02698-f003:**
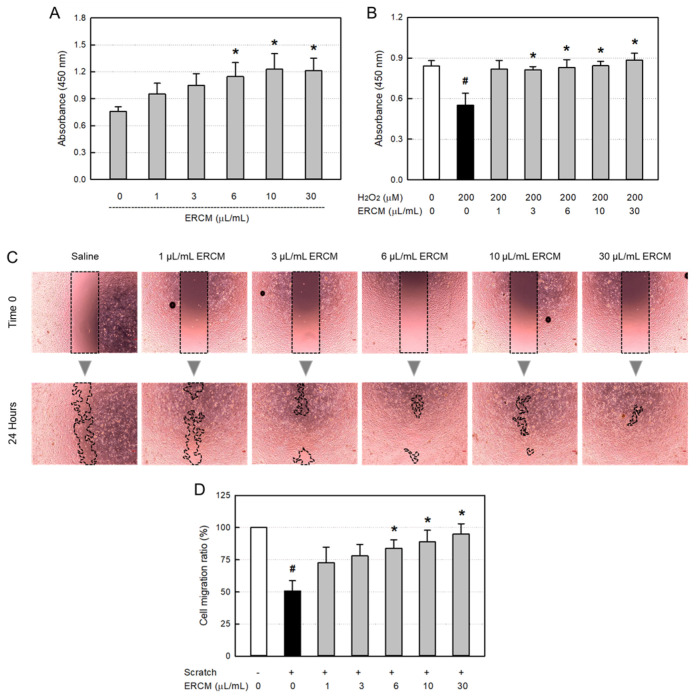
HaCaT keratinocyte-proliferative and -protective activities of exosome-rich conditioned medium (ERCM). (**A**) HaCaT cell proliferation by ERCM. (**B**) HaCaT cell protection by ERCM against oxidative stress (200 μM H_2_O_2_). The cells (1 × 10^4^/mL, *n* = 5/group) were treated with 200 μM H_2_O_2_ and ERCM (1–30 μL/mL), and incubated for 24 h. (**C**,**D**) Facilitation of HaCaT cell migration (scratch-healing) by ERCM. The cell culture plates (1 × 10^5^/mL, *n* = 5/group) were scratched, and incubated with ERCM (1–30 μL/mL) for 24 h. ^#^ Significantly different from Normal control (*p* < 0.05). * Significantly different from Scratch alone (*p* < 0.05).

**Figure 4 cells-12-02698-f004:**
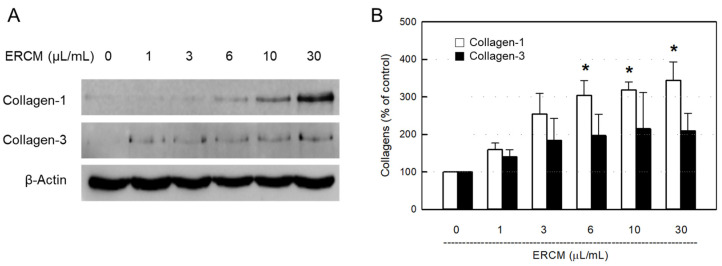
Collagen synthesis-promoting activity of exosome-rich conditioned medium (ERCM). Western blot (**A**) and quantitative (**B**) analysis of collagen–1 (white bars) and collagen–3 (black bars) in 3T3–L1 fibroblasts. The cells (1 × 10^4^/mL, *n* = 5/group) were treated with ERCM (1–30 μL/mL), and incubated for 24 h. * Significantly different from Normal control (*p* < 0.05).

**Figure 5 cells-12-02698-f005:**
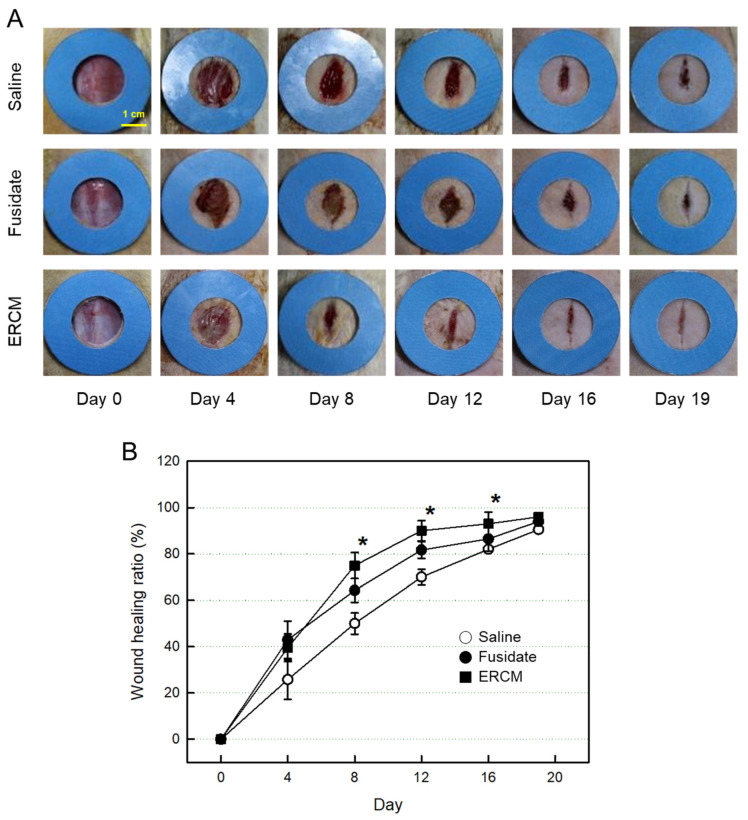
Gross findings of wound healing in rats dermally applied with fusidate sodium or exosome-rich conditioned medium (ERCM). (**A**) Representative findings. (**B**) Quantitative analysis of wound closure. Full-thickness 20-mm skin wounds (*n* = 10/group) were applied with ERCM (filled square, 4 × 10^9^ exosome particles/100 μL), Saline solution (open circle) or 3 mg Fucidin^®^ ointment (filled circle, 100 μg as fusidate sodium) on days 0, 4, 8, and 12. * Significantly different from Saline control (*p* < 0.05).

**Figure 6 cells-12-02698-f006:**
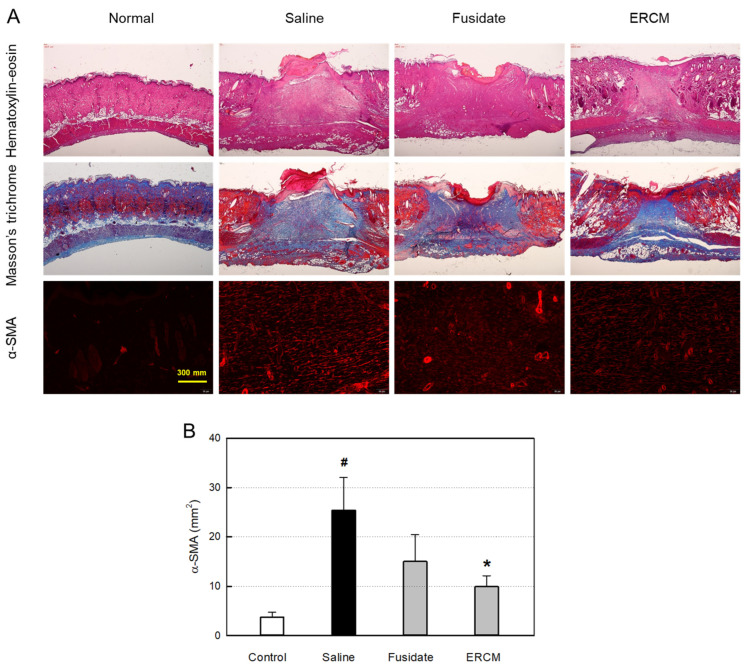
Microscopic findings of wound healing and myofibroblast differentiation in rats dermally applied with fusidate sodium or exosome-rich conditioned medium (ERCM). (**A**) Representative findings stained with hematoxylin-eosin, Masson’s trichrome or immunostained with an antibody specific for α-smooth muscle actin (α-SMA). (**B**) Quantitative analysis of α-SMA-positive cells (*n* = 5/group). ^#^ Significantly different from Normal control (*p* < 0.05). * Significantly different from Saline (Wound alone) (*p* < 0.05).

**Figure 7 cells-12-02698-f007:**
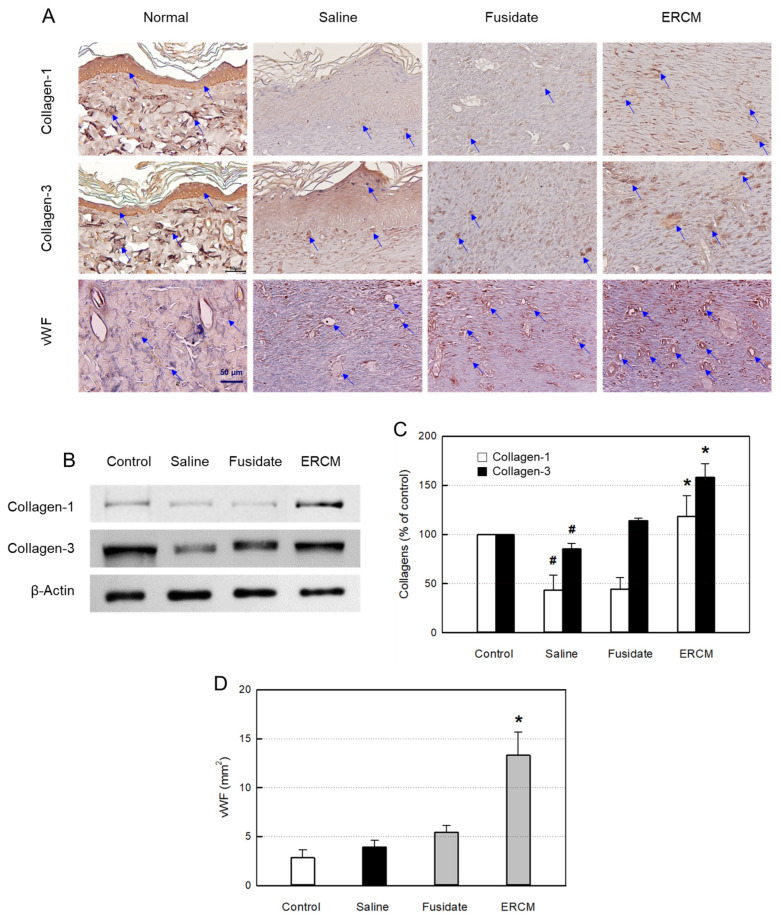
Collagen synthesis and neovascularization in the skin of rats dermally applied with fusidate sodium or exosome-rich conditioned medium (ERCM). (**A**) Representative findings immunostained with antibodies specific for collagen–1, collagen–3 or von Willebrand Factor (vWF). Arrow: vWF-positive vessel. (**B**,**C**) Western blot analysis of collagen–1 (white bars) and collagen–3 (black bars) (*n* = 5/group). (**D**) Quantitative analysis of vWF-positive cells (*n* = 5/group). ^#^ Significantly different from Normal control (*p* < 0.05). * Significantly different from Saline (Wound alone) (*p* < 0.05).

## Data Availability

Data are contained within the article.

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
