# Peer review of "An Exosome-Rich Conditioned Medium from Human Amniotic Membrane Stem Cells Facilitates Wound Healing via Increased Reepithelization, Collagen Synthesis, and Angiogenesis"

_cells, 2023, doi:10.3390/cells12232698_

Round 1

Reviewer 1 Report

Comments and Suggestions for Authors

The objective of this manuscript is to delve into the impact of Exosome-Rich Conditioned Medium (ERCM) from Human Amniotic Membrane Stem Cells on skin wound healing. The introduction is well-crafted, and the results substantiate their conclusions, making it an intriguing piece of work. However, enhancing the paper further is possible by incorporating additional experiment details in figure captions and ensuring clear labeling of columns. Here are my specific comments:

  1. Introduction:

The author is encouraged to expand the last paragraph of the introduction to elucidate the rationale behind choosing AMSCs as the source of ERCM. Consider addressing any differences in the composition of ERCM from ADSC and AMSC.

Fig 1:

  • In Fig 1A, the methods for transmission electron microscopic and Nanoparticle-Tracking Analysis are absent in the Materials and Methods section. I suggest reconsidering the use of TEM for exosome characterization, as it may alter their structure. Cryo-TEM could be a more accurate alternative. The Nanoparticle-Tracking Analysis system should be explained in the methods.

  • For Fig 1D, add a bar label in D to denote the groups of white and black columns, not just in the caption but also visually.

Fig 2:

  • Include a quantification of Figure 2. The caption lacks essential details, such as the duration of exposure to H2O2, the number of replicates, and the concentration of exosomes for treatment. Additionally, Fig 2B is missing in the caption.

Fig 3:

  • Similar to the above, provide essential details on how the experiment in Fig 3 is executed.

Fig 4:

  • Include the time of treatment and the number of replicates in Fig 4.

Fig 5:

  • Add a bar label in Fig 5B, along with any missing essential details.

Fig 6:

  • Suggest replacing low-resolution images in Fig 6 with higher resolution ones. Ensure the inclusion of information on the time of treatment and the number of replicates.

Fig 7:

  • Address the same issue as above; add a scale bar to the images in Fig 7. Clarify if the morphology differences in the normal group are due to a distinct magnification.

Comments on the Quality of English Language

NA.

Author Response

Reviewer 1:

The objective of this manuscript is to delve into the impact of Exosome-Rich Conditioned Medium (ERCM) from Human Amniotic Membrane Stem Cells on skin wound healing. The introduction is well-crafted, and the results substantiate their conclusions, making it an intriguing piece of work. However, enhancing the paper further is possible by incorporating additional experiment details in figure captions and ensuring clear labeling of columns. Here are my specific comments:

Introduction:

The author is encouraged to expand the last paragraph of the introduction to elucidate the rationale behind choosing AMSCs as the source of ERCM. Consider addressing any differences in the composition of ERCM from ADSC and AMSC.

→ We added the rationale choosing AMSCs without expressing HMC Class-II. Thanks!

Fig 1:

In Fig 1A, the methods for transmission electron microscopic and Nanoparticle-Tracking Analysis are absent in the Materials and Methods section. I suggest reconsidering the use of TEM for exosome characterization, as it may alter their structure. Cryo-TEM could be a more accurate alternative. The Nanoparticle-Tracking Analysis system should be explained in the methods.

→ We added the methods. Although we were not able to perform Cryo-TEM, the images and size distribution of the exosomes were homogeneous showing single peak at 72 nm. Thank you for valuable comments!

For Fig 1D, add a bar label in D to denote the groups of white and black columns, not just in the caption but also visually.

→ Wee added a bar label. Thank you!

Fig 2:

Include a quantification of Figure 2. The caption lacks essential details, such as the duration of exposure to H2O2, the number of replicates, and the concentration of exosomes for treatment. Additionally, Fig 2B is missing in the caption.

→ We added explanation on treatment condition

Fig 3:

Similar to the above, provide essential details on how the experiment in Fig 3 is executed.

→ We added explanation on treatment condition

Fig 4:

Include the time of treatment and the number of replicates in Fig 4.

→ We added explanation on treatment condition.

Fig 5:

Add a bar label in Fig 5B, along with any missing essential details.

→ We added a bar label, and explained treatment condition.

Fig 6:

Suggest replacing low-resolution images in Fig 6 with higher resolution ones. Ensure the inclusion of information on the time of treatment and the number of replicates.

→ We made the images a little bit bright. Actually, the images are not so bad. However, I am sorry the figures become dimmed during PDF transformation in my computer. It is expected that the Editorial office can improve the images. We also added the number of replicates.

Fig 7:

Address the same issue as above; add a scale bar to the images in Fig 7. Clarify if the morphology differences in the normal group are due to a distinct magnification.

→ We added a scale bar and the number of replicates. Actually, regenerating tissues show somewhat different features from normal tissue. It is expected that the tissues should restore original features after remodeling process.

Reviewer 2 Report

Comments and Suggestions for Authors

Supportive findings were presented in this manuscript.  Some minor suggestions: 

- Include the sample size (replicates) for all experiments.

- Fig 2A &B - legend description missing for Fig 2B. Consider including the single fluorochrome images of CM-Dil exosome to show the amount of exosomes in damaged and undamaged cells.  

Comments on the Quality of English Language

Can be improved.

Author Response

Reviewer 2:

Supportive findings were presented in this manuscript. Some minor suggestions:

- Include the sample size (replicates) for all experiments.

→ We added the sample size in each Figure legend.

- Fig 2A &B - legend description missing for Fig 2B. Consider including the single fluorochrome images of CM-Dil exosome to show the amount of exosomes in damaged and undamaged cells.

→ We corrected. Thank you! As you know, the exosomes are nanoparticles (mean size 72 nm). So, unfortunately, it was practically impossible to quantify the exosomes in cells, since we cannot extract or count them.